# Molecular Epidemiology of *Pasteurella multocida* Associated with Bovine Respiratory Disease Outbreaks

**DOI:** 10.3390/ani13010075

**Published:** 2022-12-24

**Authors:** Johan Manuel Calderón Bernal, Ana Fernández, José Luis Arnal, Celia Sanz Tejero, José Francisco Fernández-Garayzábal, Ana I. Vela, Dolores Cid

**Affiliations:** 1Departamento de Sanidad Animal, Facultad de Veterinaria, Universidad Complutense, 28040 Madrid, Spain; 2Exopol. Veterinary Diagnostic and Autogenous Vaccine Laboratory, Polígono Río Gállego, D/14. San Mateo de Gállego, 50840 Zaragoza, Spain; 3Centro de Vigilancia Sanitaria Veterinaria (VISAVET), Universidad Complutense, 28040 Madrid, Spain

**Keywords:** bovine respiratory disease, *Pasteurella multocida*, outbreaks, MLST, PFGE, virulotyping, cattle

## Abstract

**Simple Summary:**

*Pasteurella multocida* is a pathogen with increasing clinical significance in bovine respiratory disease (BRD). However, the number of studies focused on the characterization of BRD-associated *P. multocida* isolates is low. In this study, 170 *P. multocida* isolates from 125 BRD outbreaks were characterized by different typing approaches (capsular and LPS typing as well as virulotyping). Additionally, a subset of isolates was further characterized by Multilocus Sequence Typing (MLST) and Pulsed-field gel electrophoresis (PFGE). Overall, the results revealed a very low genetic diversity among *P. multocida*. These results support the clonal population structure of BRD-associated *P. multocida* isolates and corroborate the genetic relatedness of most *P. multocida* isolates associated with BRD in cattle.

**Abstract:**

Studies that characterize bovine respiratory disease (BRD)-associated *Pasteurella multocida* isolates are scarce compared with research on isolates from other hosts and clinical backgrounds. In the present study, 170 *P. multocida* isolates from 125 BRD outbreaks were characterized by capsular and LPS typing as well as by virulotyping. Three capsular types (A, B, F) and three LPS genotypes (L2, L3, L6) were identified. Capsular and LPS typing revealed a very low genetic diversity (GD = 0.02) among *P. multocida*, with most isolates belonging to genotype A:L3 (97.6%). Virulotyping identified seven virulence-associated gene profiles, with two profiles including 95.9% of the isolates. A subset of isolates was further characterized by MLST and PFGE. The sequence types ST79 and ST13 were the most frequently identified and were grouped into the same clonal complex (CC13), a result that supports the clonal population structure of BRD-associated *P. multocida* isolates. PFGE typing also revealed a low genetic diversity (GD = 0.18), detecting a single pattern in 62.5% of the outbreaks in which multiple isolates were analyzed. Overall, 85.2% of the isolates belonged to pulsotypes with at least 80% genetic similarity, consistent with a clonal population structure observed by MLST analysis and corroborating the genetic relatedness of most *P. multocida* isolates associated with BRD in cattle.

## 1. Introduction

Bovine respiratory disease (BRD) is one of the main infectious diseases in the cattle industry causing significant economic loss in fattening operations due to increased mortality, high treatment costs, or decreased growth rate [1]. BRD is a multifactorial and polymicrobial disease in which different management and environmental factors that depress host immunity interact with infectious etiological agents causing the clinical disease, usually in the form of outbreaks [2,3,4]. Different viral and bacterial pathogens are involved in the etiology of BRD [5,6,7]. *Pasteurella multocida* is one of the bacterial pathogens most frequently detected in calves affected by BRD [2,8] and has been reported with an increased incidence in the past few years in BRD outbreaks [2,8,9,10]. Moreover, *P. multocida* is also considered the main agent involved in BRD outbreaks in cattle vaccinated against other pathogens associated with BRD [11], all of which confirms the relevance of *P. multocida* in BRD.

Typing of isolates is an essential tool for understanding the epidemiology of *P. multocida* infections. Traditionally, detection of the capsule and of cell envelope lipopolysaccharides (LPSs) has been used in epidemiological studies to determine the circulating serovars of *P. multocida* [12]. Although both serotyping methods have provided useful epidemiological information, they are difficult to use in the clinical laboratory routine because of high-quality sera requirements [13,14]; therefore, these methods have recently been replaced by more accurate and rapid molecular techniques. Thus, capsular and LPS genotyping, based on the detection of capsular biosynthesis genes and the genetic organization of the LPS outer core biosynthesis loci, respectively, can classify *P. multocida* isolates into five (A, B, D, E, and F) capsular types and eight LPS (L1 to L8) genotypes that are easily identifiable by species-specific polymerase chain reaction (PCR) assays [13,15]. *P. multocida* is a genotypically diverse bacteria [16], and consequently both typing schemes have a limited capacity to differentiate related strains because they are not accurate indicators of genetic diversity between isolates [14]. Thus, in addition to capsular and LPS genotyping, other molecular typing methods such as multilocus sequence typing (MLST), pulsed-field gel electrophoresis (PFGE), and virulence genotyping based on the detection of different virulence gene profiles have been used to determine the genetic characteristics of *P. multocida* populations [16,17]. *P. multocida* isolates associated with different diseases and hosts have been characterized by a variety of these molecular typing methods [14,16,18,19,20,21,22,23,24,25,26,27]. Regarding bovine isolates, most studies have focused on the characterization of *P. multocida* isolates from hemorrhagic septicemia [16,28,29,30], while the number of studies that have included isolates from BRD is more limited [20,31,32]. Spain is one of the main beef cattle producers in the European Union (https://ec.europa.eu/eurostat/, 17 November 2022), but studies investigating the characteristics of *P. multocida* isolates in beef cattle are missing. The present study was designed to perform genetic characterization of a large collection of *P. multocida* isolates obtained from BRD outbreaks in feedlots from various geographical locations in Spain.

## 2. Materials and Methods

### 2.1. Pasteurella Multocida Isolates

A total of 170 isolates of *P. multocida* from 125 BRD outbreaks received for diagnosis at the Exopol Veterinary Diagnostic Laboratory (Zaragoza, Spain) between January 2020 and February 2022 were included in this study (Appendix A). Outbreaks occurred in 109 feedlot farms in 27 provinces of Spain (Appendix A). A total of 109 outbreaks were represented by one unique isolate (*n* = 109 isolates), 14 outbreaks by four isolates (*n* = 56 isolates), one outbreak by three isolates (*n* = 3 isolates), and one outbreak by two isolates (*n* = 2 isolates). Isolates were obtained from bronchoalveolar lavages (*n* = 114), lungs (*n* = 36), nasopharyngeal swabs (*n* = 18), and tracheal scrapes (*n* = 2). The bacteria were grown on Columbia sheep blood agar (BioMérieux España S.A., Madrid, Spain.) and incubated aerobically at 37 °C for 24 h. The identification of the isolates was confirmed by a species-specific PCR assay [33] targeting the *kmt1* gene (Table 1). The bacteria were frozen at −20 °C until further processing.

### 2.2. DNA Samples 

Bacterial DNA was obtained by suspending two to four colonies of overnight cultures in 200 µL of filtered sterile water (Sigma-Aldrich, Massachusetts, MA, USA), which was boiled at 100 °C for 10 min with stirring at 600 rpm in a thermoblock (Thermomixer comfort™ T3442, Eppendorf, Darmstadt, Germany). DNA samples were stored at −40 °C until use. The primer pairs used for amplification of the different capsular, LPS, and virulence-associated genes are listed in Table 1. Primers were synthesized by the STAB Vida laboratories (Caparica, Portugal).

### 2.3. Capsular and Lipopolysaccharide Typing 

Capsular types (A, B, D, E, and F) were determined by a multiplex PCR, as described by Townsend et al. [15]. LPS genotypes (L1 to L8) were identified by multiplex PCR, as described by Harper et al. [13]. 

### 2.4. Virulence-Associated Gene Typing 

A total of nine genes associated with virulence in *P. multocida* (*sodA*, *ompH*, *tbpA*, *tadD*, *hsf1*, *hgbB*, *pfhA*, *ptfA*, and *toxA*) were investigated. Three multiplex PCR reactions were established to detect the virulence factors *sodA*, *ompH*, and *tbpA* (reaction 1), virulence factors *tadD*, *hsf1*, and *hgbB* (reaction 2), and virulence factors *pfhA* and *ptfA* (reaction 3). The *toxA* gene was detected by simplex PCR reaction as previously described [18]. The PCR conditions, adapted from Gharib Mombeni et al. [37], were as follows: initial denaturation at 94 °C for 15 min; 25 cycles at 94 °C for 30 s, at 53 °C for 30 s, and at 72 °C for 60 s; with a final extension at 72 °C for 10 min. All reactions were performed using an MJ Mini Gradient Thermal Cycler (Bio-Rad, England). Simplex Taq PCR and Multiplex PCR Master Mix kits from Qiagen (Germantown, USA) were used in the reactions. PCR-generated products were detected by electrophoresis in 2.0% agarose gels supplemented with 1X SYBR safe DNA gel stain (Invitrogen SA) using a PowerPac basic power supply (Bio-Rad) at 90 V and 400 mA for 45 min. *P. multocida* strains NCTC 10322 (*cap-A*, *tadD*, *pfhA*, *ptfA*), NCTC 10323 (*cap-B*, *tbpA*, *hgbB*), NCTC 12178 (*cap-D*, *lps-6*, *ompH*, *toxA*), NCTC 10326 (*cap-E*, *lps-2*, *sodA*), and C104013 (*cap-F*, *lps-3*, *hsf-1*) were used as positive controls for the genes indicated.

### 2.5. Multilocus Sequence Typing (MLST)

A total of 48 *P. multocida* isolates were characterized by MLST using the *P. multocida* RIRDC database (https://pubmlst.org/organisms/pasteurella-multocida). Isolates were selected based on the different genotypes obtained in the combination of capsular–LPS and virulotypes identified in the different provinces where BRD outbreaks occurred. PCR amplification for seven housekeeping genes (*adk*, *est*, *pmi*, *zwf*, *mdh*, *gdh*, and *pgi*) was carried out using primers and protocols described by Subaaharan et al. [36] and shown in Table 1. The PCR products were purified using a commercial kit (Geneclean® Turbo Kit, MP Biomedicals, Spain), and sequenced on both strands at the STAB Vida laboratories (Caparica, Portugal) using the Sanger sequencing method. The sequence of each locus was checked on the website of the *P. multocida* RIRDC MLST database to determine the allele designations and sequence type (ST) of each isolate. STs were assigned to clonal complexes (CC) according to the *P. multocida* RIRDC MLST database.

### 2.6. Pulsed-Field Gel Electrophoresis (PFGE)

A total of 61 isolates of *P. multocida* belonging to the 16 outbreaks with multiple isolates were analyzed by PFGE. A single colony of *P. multocida* was grown in 3 mL of brain heart infusion broth (BHI) (BD Bacto, France) and incubated aerobically at 37 °C with agitation (200 rpm) for 18–22 h. Bacterial suspensions were centrifuged (12,000 rpm for 5 min at 4 °C), then washed twice with TE 1x buffer solution (1 M Tris HCl, 0.5 M EDTA, pH = 8) and finally resuspended in 400 μL of TE 1x buffer solution. Subsequently, 200 μL of this suspension was mixed with 10 μL of proteinase K (20 mg/mL) (Roche Diagnostics GmbH, Germany) and incubated at 37°C for 30 min. This suspension was mixed with an equal volume of SeaKem Gold (Lonza, Porriño, Pontevedra, Spain) low melting agarose to prepare plugs at a final concentration of 1% agarose. Once solidified, the agarose blocks were submerged in 5 mL of lysis buffer solution (1 M Tris HCl, 0.5 M EDTA, N-lauroylsarcosine to 10%, pH = 8) supplemented with 25 μL of proteinase K (20 mg/mL) (Roche Diagnostics GmbH, Germany) and incubated at 55 °C in a thermostatic bath with agitation (140 rpm) for 2 h. The blocks were washed twice with miliQ water followed by four washings with TE 1x buffer solution at 55 °C in a thermostatic bath with agitation (140 rpm) for 10 min. each. Restriction enzyme digestion was performed with an enzyme Bsp120l (Thermo Fisher, Lithuania) overnight at 37 °C according to the conditions recommended by the manufacturer. The digested DNA was loaded on 1% agarose D1 Low EEO gel (CondaLab, Spain) and separated by clamped homogeneous electric field (CHEF) electrophoresis using a CHEF-DR III System (Bio-Rad, Alcobendas, Madrid, Spain). Running conditions were as follows: initial switch time 1 s, final switching time 30 s, at a constant voltage of 6 V/cm, with a total running time of 22 h and 14°C running temperature. Gels were stained with SYBR Safe DNA (Invitrogen, CA, USA) in 0.5x TBE buffer solution. Salmonella enterica serovar Braenderup H9812 and lambda ladder PFGE marker (Boehringer Mannheim, Ingelheim am Rhein, Germany) were used to determine the molecular weight and size. Similarities between restriction endonuclease digestion profiles were analyzed visually and by using the BioNumerics software (Applied Maths BVBA, Sint-Martens-Latem, Belgium). Isolates were considered unique when they differed in at least one band. Similarities between restriction endonuclease digestion profiles of the different isolates were expressed with the Dice similarity index using the numerical taxonomy program BioNumerics (Applied Maths, BVBA, Sint-Martens-Latem, Belgium). A similarity matrix was computed and transformed into an agglomerative cluster using the unweighted pair group method with arithmetic averages (UPGMA).

### 2.7. Data Analysis

Genetic diversity (GD) was calculated as the ratio between total genotypes, virulotypes, MLST, or PFGE patterns and the total number of isolates [38]. The free online tools at http://insilico.ehu.es/mini_tools/discriminatory_power/index.php, accessed on 1 October 2022, were used to detect the discriminatory index (DI) of the different technique combinations.

## 3. Results

### 3.1. Capsular and LPS Genotypes

The capsular and LPS genotypes identified in the 170 *P. multocida* isolates are shown in Table 2. 

Capsular typing identified three capsular types (A, B, and F), with capsular type A being the most commonly detected (98.8%; *n* = 168). Capsular types B and F were detected in only one isolate each (0.6%). LPS typing identified three LPS genotypes (L2, L3, and L6): L3 was the most commonly detected genotype (98.2%; *n* = 167), whereas L2 was detected in one isolate (0.6%) and L6 in two (1.2%). When capsular types were combined with LPS genotypes (Table 2), the most commonly identified capsular–LPS genotype for the 170 *P. multocida* isolates was A:L3 (97.6 %; *n* = 166) while the other genotypes (A:L6, B:L2, and F:L3) were detected in fewer than 2% of isolates. Capsular–LPS genotypes exhibited a DI of 0.05 and a GD of 0.02.

### 3.2. Virulotypes

The frequency of detection of virulence-associated genes (VAGs) ranged between 100% and 0% of the isolates. The *sodA* gene was detected in 100% of isolates. The detection rates for the remaining VAGs were: *pfhA*, *ptfA*, *tadD* (98.8% each; *n* = 168), *tpbA* (98.2%; *n* = 167), *ompH* (14.1%; *n* = 24), hfs1 and *hgbB* (0.6%/each; *n* = 1), and *toxA* (0.0%). The 170 *P. multocida* isolates were classified into seven VAG profiles (virulotypes, VPs) based on the different combinations of the presence or absence of the VAGs analyzed (Table 2). Two profiles, VP1 and VP2, included 95.9% of the *P. multocida* isolates (VP1 = 82.9% and VP2 = 12.9%). The remaining VPs were detected in fewer than 2% of the isolates (Table 2). The virulotyping showed a DI of 0.30 and a GD of 0.04. The virulotype VP1 (84.9%) followed by VP2 (13.3%) were the most frequently detected among A:L3 genotypes (Table 2). When capsular–LPS genotypes were combined with VPs, the results showed that the capsular–LPS–VP genotype A:L3:VP1 (82.9%; *n* = 141) was the most frequently detected followed by the A:L3:VP2 genotype (12.9%; *n* = 22) (Table 2). The frequency of detection of the other capsular–LPS–VP genotypes was lower than 2% each and included fewer than 5% of the isolates (Table 2).

### 3.3. MLST

A total of 48 *P. multocida* isolates representative of the different capsular–LPS–VP genotypes identified in each province were characterized by MLST (Table 3). 

RIRDC-MLST typing assigned the 48 *P. multocida* isolates to five STs (DI = 0.53; GD = 0.10). ST79 (62.5 %) and ST13 (29.2 %) were the STs most frequently identified (Table 3). The remaining STs (ST9, ST206, and ST322) were identified in fewer than 5% of the isolates each. ST13 and ST79 belonged to the same clonal complex (ST13), while the other three STs—ST9, ST206, and ST322—belonged to ST9, ST74, and ST122 clonal complexes, respectively.

### 3.4. PFGE

The 61 *P. multocida* isolates from 16 outbreaks with multiple isolates belonged to 11 different patterns by PFGE (DI = 0.67; GD = 0.18; Figure 1). 

A single PFGE pattern was detected in 10 (62.5%) of the 16 outbreaks analyzed, whereas in six (37.5%) of the outbreaks more than one pattern was detected (Figure 1). Pattern A was detected in 12 of the 16 (75%) outbreaks, and in seven of them (58.3%) as the unique PFGE pattern detected.

## 4. Discussion

*P. multocida* is one of the most significant bacterial pathogens associated with BRD that affects the cattle-breeding industry worldwide, mainly occurring in fattening animals [7,39]. However, the number of studies dealing with the molecular characterization of *P. multocida* isolates from BRD is limited when compared with similar studies of other diseases and hosts [14,16,20,26]. In this study, we characterized *P. multocida* isolates from cases of BRD from a relatively large number of outbreaks in different regions of Spain with the highest beef cattle censuses. Genetic characteristics of *P. multocida* isolates were determined by different techniques that have proven their usefulness for the rapid and accurate characterization of *P. multocida* in epidemiological studies [17]. Thus, the results of the present study provide an accurate overview of the genetic characteristics of the *P. multocida* isolates associated with BRD circulating in beef cattle farms in Spain.

Capsular and LPS typing indicated a very low genetic diversity among *P. multocida* isolates (GD = 0.02) with only four capsular–LPS genotypes identified, of which the great majority of isolates belonged to genotype A:L3 (97.6%). This result agrees with previous works in which capsular type A or L3 genotype was commonly detected in *P. multocida* isolates associated with BRD [26,29,31,32,40]. Despite the high frequency of detection of this genotype in *P. multocida* isolates of bovine origin, it has also been frequently associated with other diseases and hosts such as avian fowl cholera or progressive atrophic rhinitis and pneumonic pasteurellosis in pigs [17] and therefore it cannot be considered to be associated only with cattle or BRD. The low DI (0.05) exhibited by the capsular type and LPS combinations indicates the limited capacity of both approaches for differentiating related strains, and shows the genetic diversity among unrelated *P. multocida* isolates. Virulotyping based on the identification of several virulence-associated genes has been successfully used for the characterization of *P. multocida* from different hosts [22,25,26,27,37] as well as for other bacterial respiratory pathogens [38,41]. Thus, *P. multocida* isolates were further characterized by virulotyping based on the presence/absence of nine VAGs that were chosen because of their variable detection rates in previous studies of bovine isolates [16,28,29,30,31,32,42] and their common use for virulence genotyping in epidemiological studies [18,21,22]. Eight of the nine genes screened were present or absent in all or in the great majority of *P. multocida* isolates, which resulted in the identification of only seven virulotypes (VP1–VP7), with VP1 and VP2 accounting for 95.9% of the isolates (82.9% and 12.9%, respectively) that resulted in a virulotyping DI of 0.30. This unexpected limited difference in the detection rates for most genes compared with the higher genetic diversity observed in previous studies of bovine isolates may be explained by the fact that these studies included not only respiratory isolates but also those from other clinical processes such as hemorrhagic septicemia and they did not differentiate the virulence genotypes according to the clinical origin of the isolates [16,28,30]. There is only one study that investigated the virulence genotypes exclusively of bovine respiratory *P. multocida* capsular type A isolates [32], and the results for the same VAGs were very similar to those obtained in this study. Although the specific virulotypes identified in this study are not easily compared with those obtained in other studies, *P. multocida* BRD isolates also demonstrated a low genetic diversity (GD = 0.04) by virulotyping, which is similar to that obtained by capsular-LPS typing.

For a global comparison with other studies, an additional analysis by MLST was performed of a subset of 48 *P. multocida* isolates, representing the different provinces in which BRD outbreaks occurred, and of the different genotypes obtained by capsular–LPS and virulotype combinations. We used the RIRDC database, which although initially created for typing of avian isolates, has been widely used for typing of *P. multocida* isolates from other species, including ruminants [19,20,22,40]. The five STs identified among the 48 isolates represent a DI of 0.53 and a GD of 0.10 consistent with the low genetic diversity observed by capsular–LPS–virulotyping combinations. Thus, more than 90% of isolates belonged to ST79 and ST13 (62.5% and 29.2%, respectively). ST79 is one of the most frequently detected sequence types in *P. multocida* bovine isolates in various European countries, in Asia, and in North America [19,20,26,40], data that confirm the worldwide distribution of this sequence type among *P. multocida* isolates from BRD. The fact that ST79 has been mainly isolated from bovine isolates, while it is not commonly detected in other hosts [16,22,23,24,25,26,40], indicates a certain host predilection of this sequence type for cattle. On the other hand, ST13 has been found in multiple hosts [16,19,22] and it is one of the most prevalent STs among porcine isolates [22,26]. In this study, most (92.3%) of the bovine ST13 isolates belonged to virulence profile VP1 that included the presence of the *tbpA* gene. The *tbpA* gene is highly prevalent among ruminant isolates and rarely found in isolates from other hosts, and therefore it has been considered an epidemiological marker for ruminant host specificity [18,22,30]. A BLAST search for sequence homologues to the *tbpA* gene in the published genomes of porcine *P. multocida* isolates, including those previously characterized as genotype A:L3:ST13 [26], did not yield positive results, indicating that bovine and porcine ST13 isolates can easily be differentiated by the presence/absence of the gene *tbpA*, respectively, and may indicate that isolates with the genotype A:L3:VP1:ST13 could be adapted to bovine hosts. Both ST79 and ST13 belonged to the same clonal complex CC13 supporting the clonal population structure of *P. multocida* isolates associated with BRD [16,19].

The genetic homogeneity in the *P. multocida* isolates from BRD observed by capsular–LPS–virulotyping–MLST combinations is consistent with the limited diversity observed in BRD-associated *P. multocida* isolates [20]. Nevertheless, to avoid a possible overestimation of this genetic homogeneity due to the fact that most outbreaks were represented by a single isolate, a subset of 61 isolates from 16 outbreaks (two–four isolates/outbreak) were characterized by PFGE to determine the genetic diversity of different isolates recovered from different animals in the same outbreak. This typing approach is especially useful in epidemiological investigations of pathogenic microorganisms due its typeability, reproducibility, and discriminatory power [43] and it has been shown to be useful in investigating the epidemiology of *P. multocida* at farm level [20]. The identification of 11 different pulsotypes also represents a low genetic diversity (GD = 0.18), consistent with that observed by the other typing methods. Moreover, 85.2% of the isolates belonged to pulsotypes that exhibited at least 80% genetic similarity (Figure 1), a result that is consistent with a clonal population structure of BRD-associated *P. multocida* isolates observed by MLST analysis and corroborates the close genetic relatedness of most *P. multocida* isolates associated with BRD in cattle. It is generally accepted that *P. multocida*, commonly present in the upper respiratory tract, is able to descend to the lungs after exposure to various risk factors [20,44]. In this sense, between two and three different pulsotypes were identified in six of the 16 outbreaks (Figure 1), which could indicate that multiple strains of *P. multocida* contribute to the pneumonic lesions and clinical signs of the disease [4,39]. However, in this study some strains in particular were consistently associated with clinical cases of BRD. Thus, pulse type A represented more than half (55.7%) of the 61 isolates and it was detected in three-quarters (75.0%) of the 16 outbreaks despite the fact that they occurred on farms located in different geographical areas (Figure 1). The high frequency of detection of pulse type A in different outbreaks and also in different animals of the same outbreak may be due to its wider distribution in cattle, to a higher prevalence in the upper respiratory tract compared to isolates of other pulsotypes, or to the existence of a specific set of genes that favor host–pathogen interactions making this pulsotype more prone to causing infection in cattle. In this respect, molecular characterization studies that enable the identification of highly prevalent BRD-associated genotypes of *P. multocida* could contribute to a more rational design of licensed vaccines designed to prevent respiratory infections caused by *P. multocida* in cattle and may improve their efficacy. 

## 5. Conclusions

In conclusion, the results of the present work confirm the low genetic diversity and the clonal population structure of *P. multocida* isolates associated with BRD. In addition, this study shows that several strains or one dominant strain of *P. multocida* can exist among different outbreaks and also within the same outbreak. 

## Figures and Tables

**Figure 1 animals-13-00075-f001:**
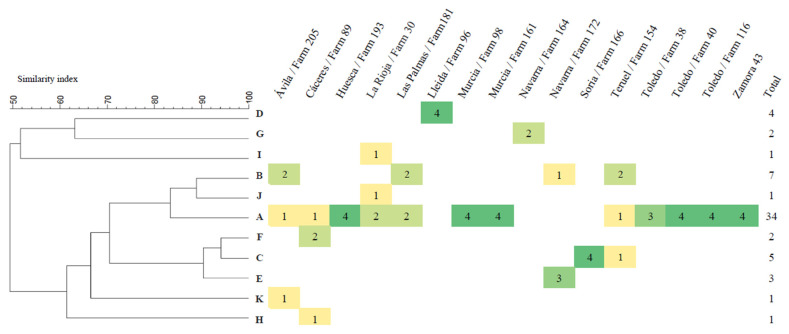
Dendrogram based on UPGMA cluster analysis showing the level of similarity observed in the 61 *P. multocida* isolates following Bsp120I endonuclease DNA digestion and heat map showing the distribution of the PFGE isolate patterns by BRD outbreak. Color scale indicates the number of isolates in each PFGE pattern from dark green (*n*= 4 isolates) to yellow (*n*= 1 isolate).

**Table 1 animals-13-00075-t001:** Primers used for molecular characterization of *Pasteurella multocida* strains.

Gene	Primer	Sequence (5′-3′)	Size (bp)	Reference
**Identification and Typing**				
KMT	kmt1T7	ATCCGCTATTTACCCAGTGG	460	[33]
	kmt1SP6	GCTGTAAACGAACTCGCCAC		
Capsular type A	hyaD/hyaC-Fw	TGCCAAAATCGCAGTCAG	1044	[15]
	hyaD/hyaC-Rv	TTGCCATCATTGTCAGTG		
Capsular type B	bcbD-Fw	CATTTATCCAAGCTCCACC	758	[15]
	bcbD-Rv	GCCCGAGAGTTTCAATCC		
Capsular type D	dcbF-Fw	TTACAAAAGAAAGACTAGGAGCCC	647	[15]
	dcbF-Rv	CATCTACCCACTCAACCATATCAG		
Capsular type E	ecbJ-Fw	TCCGCAGAAAATTATTGACTC	512	[15]
	ecbJ-Rv	GCTTGCTGCTTGATTTTGTC		
Capsular type F	fcbD-Fw	AATCGGAGAACGCAGAAATCAG	852	[15]
	fcbD-Rv	TTCCGCCGTCAATTACTCTG		
LPS genotype 1	BAP6119	ACATTCCAGATAATACACCCG	1307	[13]
	BAP6120	ATTGGAGCACCTAGTAACCC		
LPS genotype 2	BAP6121	CTTAAAGTAACACTCGCTATTGC	810	[13]
	BAP6122	TTTGATTTCCCTTGGGATAGC		
LPS genotype 3	BAP7213	TGCAGGCGAGAGTTGATAAACCATC	474	[13]
	BAP7214	CAAAGATTGGTTCCAAATCTGAATGGA		
LPS genotype 4	BAP6125	TTTCCATAGATTAGCAATGCCG	550	[13]
	BAP6126	CTTTATTTGGTCTTTATATATACC		
LPS genotype 5	BAP6129	AGATTGCATGGCGAAATGGC	1175	[13]
	BAP6130	CAATCCTCGTAAGACCCCC		
LPS genotype 6	BAP7292	TCTTTATAATTATACTCTCCCAAGG	668	[13]
	BAP7293	AATGAAGGTTTAAAAGAGATAGCTGGAG		
LPS genotype 7	BAP6127	CCTATATTTATATCTCCTCCCC	931	[13]
	BAP6128	CTAATATATAAACCATCCAACGC		
LPS genotype 8	BAP6133	GAGAGTTACAAAAATGATCGGC	255	[13]
	BAP6134	TCCTGGTTCATATATAGGTAGG		
**Virulence Associated Genes (VAGs)**				
Superoxide dismutase A	sodA-Fw	TACCAGAATTAGGCTACGC	361	[18]
	sodA-Rv	GAAACGGGTTGCTGCCGCT		
Outer membrane protein H	ompH-Fw	CGCGTATGAAGGTTTAGGT	438	[18]
	ompH-Rv	TTTAGATTGTGCGTAGTCAAC		
Transferrin binding protein A	tbpA-Fw	TTGGTTGGAAACGGTAAAGC	728	[18]
	tbpA-Rv	TAACGTGTACGGAAAAGCCC		
Putative tight adherence protein D	tadD-Fw	TCTACCCATTCTCAGCAAGGC	416	[34]
	tadD-Rv	ATCATTTCGGGCATTCACC		
Autotransporter adhesion	hsf1-Fw	TTGAGTCGGCTGTAGAGTTCG	654	[34]
	hsf1-Rv	ACTCTTTAGCAGTGGGGACAACCTC		
B hemoglobin-binding protein	hgbB-Fw	ACCGCGTTGGAATTATGATTG	788	[18]
	hgbB-Rv	CATTGAGTACGGCTTGACAT		
Filamentous hemagglutinin	pfhA-Fw	AGCTGATCAAGTGGTGAAC	275	[18]
	pfhA-Rv	TGGTACATTGGTGAATGCTG		
Type 4 fimbriae	ptfA-Fw	TGTGGAATTCAGCATTTTAGTGTGTC	488	[35]
	ptfA-Rv	TCATGAATTCTTATGCGCAAAATCCTGCTGG		
Dermonecrotic toxin	toxA-Fw	CTTAGATGAGCGACAAGGTT	865	[18]
	toxA-Rv	GGAATGCCACACCTCTATA		
**Multi-Locus Sequence Typing – RIRDC scheme**				
Adenylate kinase	adk-Fw	TTTTTCGTCCCGTCTAAGC	570	[36]
	adk-Rv	GGGGAAAGGGACACAAGC		
Esterase	est-Fw	TCTGGCAAAAGATGTTGTCG	641	[36]
	est-Rv	CCAAATTCTTGGTTGGTTGG		
Mannose-6-phosphate isomerase	pmi-Fw	TGCCTTGAGACAGGGTAAGC	739	[36]
	pmi-Rv	GCCTTAACAAGTCCCATTCG		
Glucose-6-phosphate dehydrogenase	zwf1-Fw	AATCGGTCGTTTGACTGAGC	808	[36]
	zwf1-Rv	TGCTTCACCTTCAACTGTGC		
Malate dehydrogenase	mdh-Fw	ATTTCGGGATCAGGGTTAGC	620	[36]
	mdh-Rv	GGAAAACCGGTAATGGAAGG		
Glutamate dehydrogenase	gdh-Fw	ATCGACTTCTTCCGCAGACC	702	[36]
	gdh-Rv	GCGGGTGATATTGGTGTAGG		
Phosphor glucose isomerase	pgi-Fw	ACCACGCTATTTTTGGTTGC	784	[36]
	pgi-Rv	ATGGCACAACCTCTTTCACC		

**Table 2 animals-13-00075-t002:** Virulence profiles (VP) of 170 clinical BRD *P. multocida* isolates by capsular-LPS genotypes.

Virulence Profile (VP)	Virulence Associated Genes ^1^							No. (%) of Isolates of Capsular Type and LipopolysacCharide (LPS) Genotype				
	*pfhA*	*ptfaA*	*tadD*	*tbpA*	*ompH*	*hfs1*	*hgbB*	A:L3	A:L6	B:L2	F:L3	Total
								*n* = 166	*n* = 2	*n* = 1	*n* = 1	*n* = 170
VP1	+	+	+	+	-	-	-	141 (84.9)				141 (82.9)
VP2	+	+	+	+	+	-	-	22 (13.3)				22 (12.9)
VP3	-	+	+	+	-	-	-		2 (100)			2 (1.2)
VP4	+	-	+	+	-	-	-	2 (1.2)				2 (1.2)
VP5	+	+	-	-	-	-	-			1 (100)		1 (0.6)
VP6	+	+	-	-	+	+	+				1 (100)	1 (0.6)
VP7	+	+	+	-	+	-	-	1 (0.6)				1 (0.6)

^1^ All the isolates (100%) were *sodA* positive and *toxA* negative.

**Table 3 animals-13-00075-t003:** Sequence types identified in the 48 BRD *P. multocida* isolates characterized by MLST.

Capsular and LPS Genotype	Virulence Profile	Isolates Analyzed ^1^	No. (%) of Isolates of Sequence Type (ST)				
			ST79	ST13	ST206	ST9	ST322
A:L3	VP1	30	17 (56.7)	13 (43.3)			
	VP2	12	12 (100)				
	VP4	1	1 (100)				
	VP7	1		1 (100)			
A:L6	VP3	2			2 (100)		
F:L3	VP6	1				1 (100)	
B:L2	VP5	1					1 (100)
Total		48	30 (62.5)	14 (29.2)	2 (4.2)	1 (2.1)	1 (2.1)

^1^ Isolates were representative of the different genotypes obtained in the combination of capsular-LPS and virulotypes identified in the different provinces where BRD outbreaks occurred.

## Data Availability

Not applicable.

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
