# Peer review of "Molecular Epidemiology of Pasteurella multocida Associated with Bovine Respiratory Disease Outbreaks"

_animals, 2022, doi:10.3390/ani13010075_

Round 1

Reviewer 1 Report

This is a very nicely written article and I enjoyed reading it.  There are only minor comments.  It would be nice to have a link to the other data in Fig 1 (see comment below)

Line 56 wording seems a bit awkward, maybe: … difficult to use in the clinical laboratory routine because of high-quality sera requirements or difficult to use in the clinical laboratory routine due to the requirement of high-quality sera

Line 207, line 212, line 219 and line 223 please write P. multocida in italics

Line 211 please have a full return to separate table and the explanation for 1 from the text of the results

Fig 1  The interesting point here would have been to link this to the capsule/LPS and virulence profile and to the ST.  It would be interesting to see how the ST are distributed in this PFGE pattern, as the PFGE has more pattern than MLST types.  This heat map and the number of isolates is fine but does not really show us how this is related to all the other measured diversity.  It would be good to have the diversity compared to the other methods or at least to the STs.

Line 243, 256, 261, 276, 281. 289, 290,312 and 318 In the discussion do not refer back to the tables.

Author Response

Reviewer 1

This is a very nicely written article and I enjoyed reading it.  There are only minor comments.  It would be nice to have a link to the other data in Fig 1 (see comment below)

Line 56 wording seems a bit awkward, maybe: … difficult to use in the clinical laboratory routine because of high-quality sera requirements or difficult to use in the clinical laboratory routine due to the requirement of high-quality sera. Text has been changed as suggested: “because of high-quality sera requirements” (line 57)

Line 207, line 212, line 219 and line 223 please write P. multocida in italics. These changes have been done.

Line 211 please have a full return to separate table and the explanation for 1 from the text of the results. It has been done.

Fig 1  The interesting point here would have been to link this to the capsule/LPS and virulence profile and to the ST.  It would be interesting to see how the ST are distributed in this PFGE pattern, as the PFGE has more pattern than MLST types.  This heat map and the number of isolates is fine but does not really show us how this is related to all the other measured diversity.  It would be good to have the diversity compared to the other methods or at least to the STs.

PFGE technique and MLST has been used with different purpose in this study as it is stated in the discussion. MLST has been used for a global comparison with other studies and therefore, it was performed in a subset of isolates “representing the different provinces in which BRD outbreaks occurred, and of the different genotypes obtained by capsular–LPS and virulotype combinations” (lines 279-281). On the other hand, PFGE was used “to determine the genetic diversity of different isolates recovered from different animals in the same outbreak” (lines 310-311) because of its higher discriminatory power. Therefore, these techniques have been used in different subset of isolates and the results can not be merged.

Line 243, 256, 261, 276, 281. 289, 290,312 and 318 In the discussion do not refer back to the tables. References to tables have been deleted in the discussion.

Reviewer 2 Report

In my opinion, methodology of the study and data analysis support the conclusions. The manuscript is properly written and suitable for publication. Nevertheless, there is still one question: The isolate from outbreak #8 was characterised as capsular-LPS-type B:L2. This type is commonly associated with haemorrhagic septicaemia (HS), a disease also reported from Spain very recently. Therefore, could HS securely ruled out in that case?

Author Response

Reviewer 2

In my opinion, methodology of the study and data analysis support the conclusions. The manuscript is properly written and suitable for publication. Nevertheless, there is still one question: The isolate from outbreak #8 was characterised as capsular-LPS-type B:L2. This type is commonly associated with haemorrhagic septicaemia (HS), a disease also reported from Spain very recently. Therefore, could HS securely ruled out in that case?

We agree with the referee that the capsular-LPS-type B:L2 is associated with HS in cattle but it has been also described in HS-like porcine case in Spain (García-Alvarez et al, 2017). The genotype of our isolate was confirmed by duplicate analysis. Moreover, outbreak # 8 was sent for laboratory diagnosis based on clinical and epidemiological data compatible with BRD based on veterinary clinician observation. Hence, despite the possibility of HS could not be completely ruled out, this option seems to be very unlikely.

Reviewer 3 Report

Thank you for the opportunity to review this manuscript.  I have a few comments and questions for the authors to consider:

Lines 45 and 46:  I think this depends on the population being studied and management history of animals within a population.  In dairy calves this is certainly true. In beef cattle, there is quite a bit of variation seen.  Generally speaking, M. haemolytica is more common in populations like the one being studied here.

Materials and methods

Having more information on the origin of the isolates being studied is important and would enhance the manuscript.  It seems as though most of the isolates come from feedlots but that isn't clear and that point could be clarified further.  I also think classifying the genotype of isolates with respect to farm of origin would be informative so as to show the reader how much diversity exists between isolates collected from different operations.  This is done for PFGE but not for the other methods.  For consistency this is important

What was the management history of the animals on the farm of interest?  Is that information known?  If so, vaccination history, history of antimicrobial exposure, etc, are important to know as those things can influence the population of organisms in the upper airway

I'm not sure comparing isolates collected by different sampling techniques is appropriate, as it adds another variable that is not accounted for in the analysis.  I would suggest using the sampling technique (BAL, lung, NPS) as another covariate in the analysis, or just focusing on isolates collected by one sampling method. 

Please define DI and how it is calculated.  What information does it tell you, specifically?

Figure 1:  Having a vertical line representing the cut off for genetic similarity would be helpful for the reader.  Along those same lines, at what level of similarity (80%, 90%, etc) do the authors consider the isolates the same?  In the discussion it looks like 80% is being used.  This is an unusual cut point as 90% is far more common in most situations.

Discussion:  It seems as though the results of MLST disagree with the other typing methods given that the DI is 0.53 and for others it's 0.7 or lower.  Since no statistical analysis is performed, it's impossible to know if these numbers are the same, differ by random chance, or are truly different.  For example, the GD of PFGE is 0.18, while the GD of MLST is 0.10.  This is nearly double.  Some further analysis is essential to determine what this difference really means.  11 different pulsotypes is different from just 5 STs found by MLST, suggesting that PFGE might have some of the same limitations that the other methods do

Author Response

Reviewer 3

Thank you for the opportunity to review this manuscript.  I have a few comments and questions for the authors to consider:

 Lines 45 and 46:  I think this depends on the population being studied and management history of animals within a population.  In dairy calves this is certainly true. In beef cattle, there is quite a bit of variation seen.  Generally speaking, M. haemolytica is more common in populations like the one being studied here. It has been changed to “Pasteurella multocida is one of the bacterial pathogens most frequently…” (line 45).

Materials and methods

Having more information on the origin of the isolates being studied is important and would enhance the manuscript.  It seems as though most of the isolates come from feedlots but that isn't clear and that point could be clarified further.  I also think classifying the genotype of isolates with respect to farm of origin would be informative so as to show the reader how much diversity exists between isolates collected from different operations. This is done for PFGE but not for the other methods.  For consistency this is important

All the outbreaks were from feedlots. This information has been included in material and methods: “Outbreaks occurred in 109 different feedlot farms in 27 provinces of Spain (Supplementary Table).” (lines 84-85).

Data about distribution of capsular-LPS genotypes, virulotypes and STs by outbreaks and farms is included in Supplementary Table S1. The diversity between different operations as between outbreaks was very limited since most of the isolates (82.9 %) belonged to A:L3 genotype and VT1 virulotype (Table 2). On the other hand, despite most outbreaks were represented by single isolates, in some outbreaks several isolates from different animals in the same outbreak were available. Isolates of these outbreaks were characterized by PFGE as this typing approach, due its discriminatory power, is useful in investigating the epidemiology of bacterial pathogens at farm level. This is the reason why data related with the farm origin of the isolates is only indicated for PFGE results, while data of the other typing techniques are focused in a more global comparison of the P. multocida isolates from cases of BRD from outbreaks in different regions of Spain in order to have an accurate overview of the genetic characteristics of the P. multocida isolates associated with BRD circulating in beef cattle farms in Spain. Therefore, as the objectives of the PFGE and the other typing method are not the same, we do not believe that it is essential that their results must be presented in the same format. We hope you can understand and accept this point of view.

 What was the management history of the animals on the farm of interest?  Is that information known?  If so, vaccination history, history of antimicrobial exposure, etc, are important to know as those things can influence the population of organisms in the upper airway

We do not have specific data about management history in the different farms, but in all cases cattle are managed under intensive farming conditions.  The individual vaccination history of each farm could not be recorded in this study, neither history of antimicrobial exposure. Nevertheless, all the samples were collected from animals previously to antimicrobial treatment. Regarding vaccination to prevent BRD, it is a very common practice in Spanish feedlots at the entrance of animals which is performed with commercial vaccines that normally includes other etiological agents involved in BRD as bovine respiratory virus, parainfluenza 3 virus and Mannheimia haemolytica A1 but they do not include P. multocida. Thus, it was expected that nearly all animals were vaccinated.

I'm not sure comparing isolates collected by different sampling techniques is appropriate, as it adds another variable that is not accounted for in the analysis.  I would suggest using the sampling technique (BAL, lung, NPS) as another covariate in the analysis, or just focusing on isolates collected by one sampling method. 

The sampling techniques used in this study have been previously shown to be suitable to perform laboratorial BRD diagnosis (Pardon and Buczinski, Vet Clin North Am Food Anim Pract. 2020 36:425-444. Bovine Respiratory Disease Diagnosis: What Progress Has Been Made in Infectious Diagnosis?). Analysis of diversity according to the kind of sample was done, but differences were not found, which is not unexpected because of the low genetic diversity found.

Please define DI and how it is calculated.  What information does it tell you, specifically?

The discriminatory index (DI) is the average probability that the typing system will assign a different type to two unrelated strains randomly sampled in the microbial population of a given taxon. This definition is included in the web page which is given as reference in material and methods. DI is used to shown the suitability of the technique used to type the population of interest.

Figure 1:  Having a vertical line representing the cut off for genetic similarity would be helpful for the reader.  Along those same lines, at what level of similarity (80%, 90%, etc) do the authors consider the isolates the same?  In the discussion it looks like 80% is being used.  This is an unusual cut point as 90% is far more common in most situations.

We consider isolates are the same when they exhibit identical PFGE macrorestriction pattern (pulsotype). The level of similarity is a criterium used to point out the degree of genetic relatedness of isolates with different pulsotypes. In this regard, despite the percentage of genetic similarity of 90% is frequently used, there is not a fixed value, and percentages between 80% to 90% are also frequently used in many other studies to remark the genetic relatedness of isolates [see for example, Calvez et al., Vet. Res. 46, 73 (2015); Dhaka et al., Infection, Ecology and Epidemiology 6, 1 (2016) and Tramuta et al., Austin J. Microbiol. 7, 1037 (2022)].  For this reason, we indicated that most of the P. multocida isolates exhibited at least 80% genetic similarity” to point out their close genetic relatedness.

Discussion:  It seems as though the results of MLST disagree with the other typing methods given that the DI is 0.53 and for others it's 0.7 or lower.  Since no statistical analysis is performed, it's impossible to know if these numbers are the same, differ by random chance, or are truly different.  For example, the GD of PFGE is 0.18, while the GD of MLST is 0.10.  This is nearly double.  Some further analysis is essential to determine what this difference really means.  11 different pulsotypes is different from just 5 STs found by MLST, suggesting that PFGE might have some of the same limitations that the other methods do

Different molecular typing methods have different discriminatory power. Thus, it is well known that PFGE in more discriminative than MLST. This explains the different DI of both techniques. Moreover, PGFE and MLST were used with different purposes:  MLST was used for a global comparison of isolates as indicated in discussion (lines 278-281) while PFGE was used for determining the genetic diversity of isolates within the same outbreak, due to its higher discriminatory power (lines 307-319).  Therefore, the differences between both techniques make that any statistical analysis of the differences found will be inapplicable.